# Compositional Instruction Following with Language Models and Reinforcement Learning

## Abstract

Combining reinforcement learning with language grounding is challenging as the agent needs to explore the environment for different language commands at the same time. We present a method to reduce the sample complexity of RL tasks specified with language by using compositional policy representations. We evaluate our approach in an environment requiring reward function approximation and demonstrate compositional generalization to novel tasks. Our method significantly outperforms the previous best non-compositional baseline in terms of sample complexity on 162 tasks. Our compositional model attains a success rate equal to an oracle policy's upper-bound performance of 92%. With the same number of environment steps the baseline only reaches a success rate of 80%.

## 1 Introduction

An important goal of reinforcement learning (RL) is the creation of agents capable of generalizing to novel tasks. Additionally, natural language provides an intuitive way to specify a variety of tasks. Natural language has a few important properties: it is compositional in its grammar and often mirrors the compositional structure of the tasks being solved. Previous works have attempted to use natural language to specify tasks for RL agents (Ahn et al., 2023; Blukis et al., 2020). In this work we exploit the compositional nature of language along with compositional policy representations to demonstrate improvements in sample complexity and generalization in solving novel tasks.

Approaches to map language to behaviors have previously attempted to use policies learned using imitation learning (Ahn et al., 2023; Blukis et al., 2020). We want to focus our attention to problems where the agent might not have access to supervised demonstrations. We want to use RL to learn to ground language to specified behaviors. The challenge in such an approach is that there is significantly high sample complexity of RL-based methods when grounding behaviors as agents must map a variety of potential language instructions to unknown corresponding behaviors. Pretraining and transfer learning offers one possible solution. In natural language processing, pretraining language models such as BERT (Devlin et al., 2019) and GPT-4 (OpenAI, 2023) have enabled substantial reductions in sample complexity of solving novel tasks. However, in RL there is a lack of pretrained learned policy representations that can be fine-tuned using novel examples in few-shot settings.

However, a different strategy has been exploited in policy learning where solutions to previously-learned tasks can be composed to solve novel tasks (Todorov, 2009; Nangue Tasse et al., 2020). For instance, Nangue Tasse et al. (2020) demonstrate zero-shot task solving using compositional value functions and Boolean task algebra. We attempt to exploit such compositional value functions with pretrained language models to solve a large number of tasks using RL, while not relying on curricula, demonstrations or other external aids to solve novel tasks. Leveraging compositionality is essential to solving large numbers of tasks with shared structure. The sample complexity of learning large number of tasks using RL is often prohibitive unless methods leverage compositional structure (Mendez-Mendez & Eaton, 2023).

This work builds on the Boolean compositional value function representations of Nangue Tasse et al. (2020) to construct a system for learning compositional policies for following language instructions. Our insight is that language commands reflect the compositional structure of the environment, but without compositional RL representations, this structure cannot be used effectively. Likewise, it

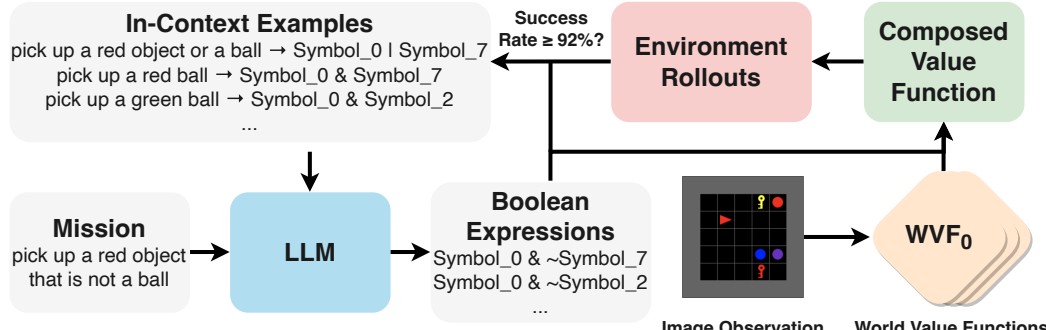

Figure 1: Pipeline diagram of the learning process for the LLM agent. The LLM agent takes in a BabyAI language mission command and a set of 10 in-context examples are selected using the BM25 search retrieval algorithm (Robertson et al., 2009). The LLM produces 10 candidate Boolean expressions. These expressions specify the composition of the base compositional value functions. Each compositional value function is instantiated in the environment and the policy it defines is evaluated over 100 rollouts. If the success rate in reaching the goal is greater than 92%, the expression is considered a valid parse of the language instruction and is added to the set of in-context examples.

is challenging to learn how to compose the RL representations without task-specific information. Language, therefore, unlocks the utility of compositional RL allowing us to not only compose base policies, but also negate their behaviors to solve tasks such as "Don't start the oven." These language-conditioned compositional RL policies can be used as pretrained general-purpose policies and novel behaviors can be added as needed when solving new tasks. Moreover, the composed policies themselves are interpretable as we can inspect the base policies that are composed.

Our primary contributions are as follows:

1. We present a novel approach for solving tasks specified using language. The policies for the tasks are represented as conjunctions, disjunctions, and negations of pretrained compositional value functions.

2. We combine in-context learning with feedback from environment rollouts to improve the semantic parsing capabilities of the LLM. As far as we are aware, our method is the first to learn a semantic parser using in-context learning with feedback from environment rollouts.

3. We solve 162 unique tasks within an augmented MiniGrid-BabyAI domain (Chevalier-Boisvert et al., 2023; 2019) which to the best of our knowledge is the largest set of simultaneously learned language-RL tasks.

4. Our method significantly outperforms the previous best non-compositional baseline in terms of sample complexity. Our compositional model attains a success rate equal to an *oracle policy's upper-bound performance of* 92%. With the same number of environment steps, the baseline only reaches a success rate of 80%.

## 2 BACKGROUND

### BABYAI DOMAIN

Because we build on the compositional value function representations of Nangue Tasse et al. (2020), our method is applicable to any environment with goal-reaching tasks, the ability to learn value functions through RL, and language instructions. To evaluate our method, we select the BabyAI MiniGrid domain (Chevalier-Boisvert et al., 2019) which provides a test-bed for compositional language-RL tasks. It has image-state observations, a discrete action space, and objects with color and type attributes. We augment BabyAI with additional Boolean compositional tasks specified using intersection, disjunction, and negation. Figure 2 provides an example of a goal-reaching present in the BabyAI domain, and its compositional specification using Boolean operators. Appendix Table 4 provides a full list of task attributes available in the environment and the grammar of the Boolean expressions. Lastly Table 1 provides examples of the types of tasks our method learns. The

environment is initialized with one or more goal objects and distractor objects that are randomly placed.

## Compositional World Value Functions (WVFs)

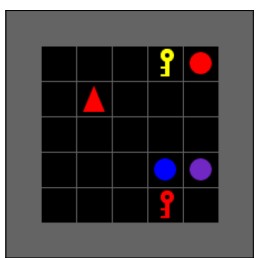

Figure 2: Example of a task in the BabyAI domain (Chevalier-Boisvert et al., 2019). The agent (red triangle) needs to complete the mission – "pick up the red key". Solving this task with compositional value functions requires using the conjunction of the *pickup* "red object" and "key" value functions.

We consider the case of an agent required to solve a series of related tasks. Each task is formalized as a Markov decision process (MDP) $\langle \mathcal{S}, \mathcal{A}, p, r \rangle$, where $\mathcal{S}$ is the state space and $\mathcal{A}$ is the set of actions available to the agent. The transition dynamics $p(s'|s,a)$ specify the probability of the agent entering state $s'$ after executing action $a$ in state $s$, while $r(s,a,s')$ is the reward for executing $a$ in $s$. We further assume that $r$ is bounded by $[r_{\text{MIN}}, r_{\text{MAX}}]$. We focus here on goal-reaching tasks, where an agent is required to reach a set of terminal goal states $\mathcal{G} \subseteq \mathcal{S}$.

Tasks are related in that they differ only in their reward functions. Specifically, we first define a background MDP $M_0 = \langle \mathcal{S}_0, \mathcal{A}_0, p_0, r_0 \rangle$. Then, any new task $\tau$ is characterized by a task-specific reward function $r_\tau$ that is non-zero only for transitions entering $g$ in $\mathcal{G}$. Consequently, the reward function for the resulting MDP is given by $r_0 + r_\tau$.

The agent aims to learn an optimal policy $\pi$, which specifies the probability of executing an action in a given state. The value function of policy $\pi$ is given by $V^\pi(s) = \mathbb{E}_\pi \left[ \sum_{t=0}^\infty r(s_t, a_t) \right]$ and represents the expected return after executing $\pi$ from $s$. Given this, the optimal policy $\pi^*$ is that which obtains the greatest expected return at each state: $V^{\pi^*}(s) = V^*(s) = \max_\pi V^\pi(s)$ for all $s \in \mathcal{S}$. Closely related is the action-value function, $Q^\pi(s,a)$, which represents the expected return obtained by executing $a$ from $s$, and thereafter following $\pi$. Similarly, the optimal action-value function is given by $Q^*(s,a) = \max_\pi Q^\pi(s,a)$ for all $(s,a) \in \mathcal{S} \times \mathcal{A}$.

## Logical Composition of Tasks using World Value Functions

Recent work (Nangue Tasse et al., 2020; 2022) has demonstrated how logical operators such as conjunction ($\wedge$), disjunction ($\vee$) and negation ($\neg$) can be applied to value functions to solve semantically meaningful tasks compositionally with no further learning. To achieve this, the reward function is extended to penalise the agent for attaining goals it did not intend to:

$$\bar{r}(s,g,a) = \begin{cases} \bar{r}_{MIN} & \text{if } g \neq s \in \mathcal{G} \\ r(s,a) & \text{otherwise,} \end{cases} \tag{1}$$

where $\bar{r}_{MIN}$ is a large negative penalty. Given $\bar{r}$, the related value function, termed *world value function (WVF)*, can be written as: $\bar{Q}(s,g,a) = \bar{r}(s,g,a) + \int_{\mathcal{S}} \bar{V}^{\bar\pi}(s',g) p(s'|s,a) ds'$, where $\bar{V}^{\bar\pi}(s,g) = \mathbb{E}_{\bar\pi} \left[ \sum_{t=0}^\infty \bar{r}(s_t, g, a_t) \right]$.

These value functions are intrinsically *compositional* since if a task can be written as the logical expression of previous tasks, then the optimal value function can be similarly derived by composing the learned WVF's. For example, consider the `PickUpObject` environment shown in Figure 2. Assume the agent has separately learned the task of collecting red objects (task $R$) and keys (task $K$). Using these value functions, the agent can immediately solve the tasks defined by their union ($R \vee K$), intersection ($R \wedge K$), and negation ($\neg R$) as follows: $\bar{Q}^*_{R \vee K} = \bar{Q}^*_R \vee \bar{Q}^*_K := \max\{\bar{Q}^*_R, \bar{Q}^*_K\}$, $\bar{Q}^*_{R \wedge K} = \bar{Q}^*_R \wedge \bar{Q}^*_K := \min\{\bar{Q}^*_R, \bar{Q}^*_K\}$, and $\bar{Q}^*_{\neg R} = \neg \bar{Q}^*_R := (\bar{Q}^*_{MAX} + \bar{Q}^*_{MIN}) - \bar{Q}^*_R$.

$\bar{Q}^*_{MAX}$ and $\bar{Q}^*_{MIN}$ are the world value functions for the *maximum* and *minimum* tasks respectively.[1]

---

[1]The maximum task is defined by the reward function $r = r_{\text{MAX}}$ for all $\mathcal{G}$. Similarly, the minimum task has reward function $r = r_{\text{MIN}}$ for all $\mathcal{G}$.

## 3 METHODS

We propose a two-step process for training an RL agent to solve the Boolean compositional tasks with language. During an initial pretraining phase, a set of WVFs are learned which can later be composed to solve new tasks in the environment. This set forms a task basis that can express any task which can be written as a Boolean algebraic expression using the WVFs.

In a second phase, an LLM learns to semantically parse language instructions into the Boolean compositions of WVFs using RL. Notably, our method does not require the semantic parser to have access to any knowledge of the underlying basis tasks that the WVFs represent, and instead regards the WVFs as abstract symbols which can be composed to solve tasks. Since the semantic parser does not have access to any information about what task a WVF represents, our method can be to be applied to any basis of tasks. The parser must therefore learn this mapping from abstract symbols to WVFs using RL by observing language instructions and interacting with the environment.

Tasks like "pickup the red key" can be represented by taking the intersection of the WVFs for picking up "$red$" objects and "$key$" objects: $red \wedge key$. Our method also supports negation and disjunction, we can specify tasks like "pick up a red object that is not a ball": $red \wedge \neg ball$. We augment this domain with additional tasks. For further examples of tasks, see Table 1, which lists the complete set of tasks created using the attributes $yellow$ and $key$. We implement the model from Chevalier-Boisvert et al. (2019) as a non-compositional baseline. This model does not have a pretraining phase for its RL representations, and in our experiments we account for this discrepancy in training steps by penalizing our agent by the number of training steps needed to learn the WVFs.

### 3.1 PRETRAINING WORLD VALUE FUNCTIONS

During pretraining, a set of WVFs is learned which can later be composed to solve any task in the BabyAI environment. Each WVF takes as input a $56 \times 56 \times 3$ RGB image observation of the environment and outputs $|\mathcal{G}| \times |\mathcal{A}| = 18 \times 7$ values for accomplishing one of the basis tasks (by maximising over the goal-action values). As there are nine object attributes (three object type attributes and six color attributes as listed in Appendix Table 4), we train nine WVFs. Each WVF is a value function for the policy of picking up objects that match one of the nine attributes. However, our method does not require knowledge of the underlying semantics of the value functions (i.e. the names of the underlying task attributes). We therefore assign each WVF a random identifier, denoted as $Symbol\_0$ through $Symbol\_8$. While we refer to the WVFs by their color and object type in the paper, our model does not have access to this information and only represents the WVFs by their unique identifiers.

Each WVF is implemented using $|\mathcal{G}| = 18$ CNN-DQN (Mnih et al., 2015) architectures. The WVF pretraining takes nineteen million steps. This is done by first training $\bar{Q}^*_{MIN}(s, g, a)$ for one million steps and $\bar{Q}^*_{MAX}(s, g, a)$ for eighteen million steps (one million steps per goal in the environment). Each basis WVF $\bar{Q}^*_B(s, g, a)$ is then generated from $\bar{Q}^*_{MIN}(s, g, a)$ and $\bar{Q}^*_{MAX}(s, g, a)$ by computing $\bar{Q}^*_B(s, g, a) = \bar{Q}^*_{MAX}(s, g, a)$ **if** $r_B(g, a) = r_{MAX}$ **else** $\bar{Q}^*_{MIN}(s, g, a)$. This yielded a 98% success rate for each basis WVF. For more details on WVF pretraining see Section 2 and Nangue Tasse et al. (2022), and see Appendix Table 6 for a full list of hyperparameters used in training the WVFs.

### 3.2 COMPOSITIONAL LLM AGENT

We assume the downstream task set is distinct from the basis task set. During downstream task learning, the pretrained WVFs are composed to solve novel tasks specified in language. To solve the BabyAI language instruction tasks, the agent must interpret the input language command and pick up an object of an allowed type. To accomplish this, the semantic parser maps from language to a Boolean expression specifying the composition of WVFs. These Boolean expressions are then composed using a fixed pipeline that takes as input the set of WVFs and the Boolean expression. This pipeline parses the Boolean expression and returns a composed WVF. The agent then acts in the environment under the policy of the WVF by taking the action with the greatest value at each step. If the Boolean expression is not syntactically correct, it cannot be instantiated as a WVF and the episode terminates unsuccessfully.

Table 1: Example language instructions and corresponding Boolean expressions for the $yellow$ and $box$ attributes

| Language Instruction | Ground Truth Boolean Expression |
| --- | --- |
| pick up a yellow box | $yellow \, \& \, box$ |
| pick up a box that is not yellow | $\sim yellow \, \& \, box$ |
| pick up a yellow object that is not a box | $yellow \, \& \, \sim box$ |
| pick up an object that is not yellow and not a box | $\sim yellow \, \& \, \sim box$ |
| pick up a box or a yellow object | $yellow \, | \, box$ |
| pick up a box or an object that is not yellow | $\sim yellow \, | \, box$ |
| pick up a yellow object or not a box | $yellow \, | \, \sim box$ |
| pick up an object that is not yellow or not a box | $\sim yellow \, | \, \sim box$ |
| pick up a box | $box$ |
| pick up an object that is not a box | $\sim box$ |
| pick up a yellow object | $yellow$ |
| pick up an object that is not yellow | $\sim yellow$ |

### 3.2.1 IN-CONTEXT SEMANTIC PARSING WITH REINFORCEMENT LEARNING

To implement the semantic parser, we utilize state-of-the-art large language models: GPT 4 (OpenAI, 2023) and GPT 3.5.[2] Our method builds on the work of Shin et al. (2021) which builds a semantic parser using LLMs and few-shot learning, and Toolformer (Schick et al., 2023) which learns an LLM semantic parser from weak supervision. Our semantic parser is distinct from these approaches in that it utilizes in-context examples combined with an environment rollout RL signal. At the start of learning, the agent has no in-context examples of valid mappings from language to Boolean expressions (see Figure 1). At each episode, the LLM is prompted with general instructions defining the semantic parsing task, the input language command, and up to 10 previously-acquired in-context examples selected using the BM25 retrieval algorithm (Robertson et al., 2009).

During training, the LLM is sampled with a temperature of 1.0 and produces a beam of 10 semantic parses (Boolean expressions) of the input language command. Together the temperature and beam width control the exploitation-exploration trade-off of the semantic parsing model. Each candidate Boolean expression is parsed using a fixed pipeline and instantiated as a WVF. The policy defined by the WVF is evaluated in the environment over 100 episode rollouts. If the success rate across these episodes in reaching the specified goals is greater than or equal to 92%, the language instruction and Boolean expression are added to the list of in-context examples. Multiple Boolean expressions may attain high reward for any given task. To counter this we add a form of length-based regularization. If the agent already has an in-context example with the same language instruction, the length of the Boolean expressions is compared and only the shorter of the two expressions is retained as an in-context example. We thereby favor shorter Boolean expressions that attain high reward in the environment. For more details of the prompting strategy, see Table 2. Hyperparameters are available in Appendix Table 5.

### 3.3 BASELINES

The baseline is a joint language and vision model which learns a single action-value function for all tasks based on the architecture used in the original BabyAI paper Chevalier-Boisvert et al. (2019). We explore two baseline models: an LM-Baseline that utilizes pretrained language representations for embedding mission commands from a frozen "all-mpnet-base-v2" model from the SentenceTransformers library Reimers & Gurevych (2019) based on the MPNet model Song et al. (2020) and an ablated Baseline which does not use pretrained language representations. This pretrained sentence embedding model is trained on diverse corpora of sentence embedding tasks.

---

[2]https://platform.openai.com/docs/models/gpt-3-5

Table 2: The prompting strategy for the LLM semantic parsing module.

| Role | Content |
|---|---|
| *System* | "We are going to map sentences to Boolean expressions. The Boolean expression variable Symbols are numbered 0 to 8, e.g. $Symbol\_0$, $Symbol\_1$... The operators are and : &, or : |, not : ~. I will now give a new sentence and you will come up with an expression. Now wait for a new sentence command. Respond with a list of 10 candidate Boolean expressions. Respond only with the list of Boolean expressions. Never say anything else." |
| *User (Example)* | "pick up a red ball" |
| *Assistant* | "$Symbol\_0 \& Symbol\_7$" |
| | [Additional in-context examples] |
| *User (Command)* | "pick up a red object that is not a ball" |
| *Assistant* | "$Symbol\_0 \& Symbol\_1 \& \sim Symbol\_2$" |
| | "$Symbol\_3 \& \sim Symbol\_4$" |
| | "$Symbol\_5 \& Symbol\_6 \& \sim Symbol\_7$" |
| | [Additional candidate expressions] |

## 4 RESULTS

We conduct experiments across four agent types and two settings. The first experiment evaluates sample complexity (Figure 3). We learn all 162 tasks simultaneously and plot the mean success rate against the number of environment steps. The second experiment divides the task set in half, and measures the ability of the agents to generalize to held-out novel tasks while learning from a fixed set of tasks (Figure 4).

We evaluate our LLM Agent implemented with GPT-4 and GPT-3.5. We compare our method to the baseline agents, but penalize our method by the number of environment steps required to learn the pretrained WVFs. As an upper limit on the performance of the LLM Agent, we also compare to an Oracle Agent which has a perfect semantic parsing module. It has access to the ground-truth mappings from language to Boolean expressions and its performance is limited only by the accuracy of the pretrained policies and randomness in the environment.

### 4.1 SIMULTANEOUS LEARNING OF 162 TASKS

In this experiment, at each episode a random task is sampled from the set of 162 language tasks. The baseline agents learn for 21 million steps, and the LLM Agents learn for 2 million steps. Because our agent pretrains the WVFs, we penalize our agent by starting it at 19 million steps (Figure 3). Note that this disproportionately disadvantages the LLM Agent, as the WVF pretraining phase does not include language information and its only exposure to language-task data is over the following two million steps. The LLM Agent therefore has access to less information about the tasks structure than the baseline agents during the first 19 million steps. For the LLM Agents, due to the latency and cost of invoking the LLM, we only evaluate on one randomly selected task every $5,000$ environment steps, computing the average performance over 100 episodes. For the baseline agents we evaluate all 162 tasks every $50,000$ timesteps. This results in higher variance for the LLM Agent methods in the plots.

We also plot the number of in-context training examples added to the LLM Agent's set in Figure 5. This is equivalent to the number of training tasks successfully solved at that step. The Oracle Agent solves the overwhelming majority of tasks during their first occurrence and is limited only by the small amount of noise in the policies and environment.

### 4.2 HELD-OUT TASK GENERALIZATION

This experiment (Figure 4) measures the generalization performance of each method on held-out tasks. We compare the performance of the GPT-4 agent to the baseline agents. In this setting, the set of tasks is randomly split into two halves at the start of training. At each episode, a random task from the first set is selected. During evaluation of the LLM Agent one random task from each set is

selected and the agent is evaluated over 100 episodes. The baseline agents are evaluated over all 81 tasks in each set.

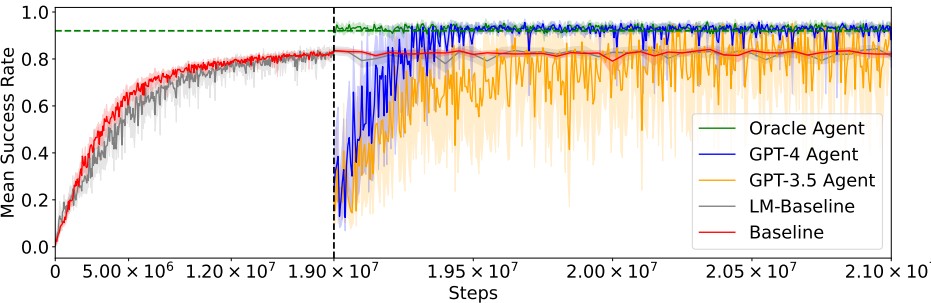

Figure 3: Results for learning all 162 tasks simultaneously. The mean episode success rate is plotted against the number of environment steps. Learning curves are presented for the LLM agent using GPT-4 and GPT 3.5 and the non-compositional baseline agents. The Oracle agent is given the ground-truth Boolean expressions and upper bounds the attainable success rate in the environment, denoted by the dashed line at $92\%$. The LLM agents are initialized at 19 million steps to reflect the number of training steps used in pretraining the compositional value functions. Note the change in steps scale at 19 million steps. Means and $95\%$ confidence intervals are reported over 10 trials, 5 trials for the LM-Baseline.

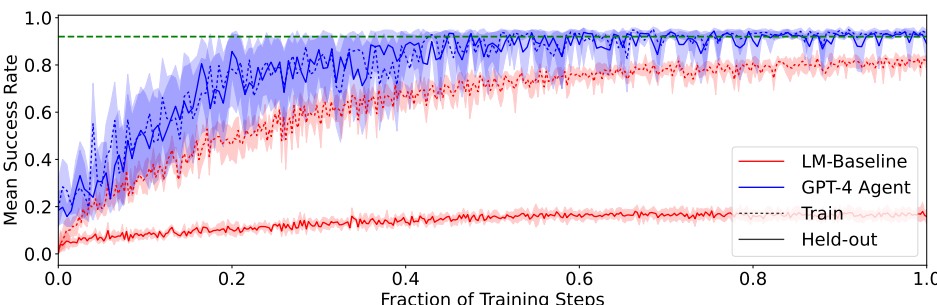

Figure 4: Results for learning on 81 randomly selected tasks and evaluating on the held-out 81 tasks. Learning curves are presented for the LLM agent using GPT-4 and the baseline agents. The $x$-axis represents the fraction of the total training steps completed: 1 million for the GPT-4 Agent and 21 million for the LM-Baseline agent. The dashed line denotes the success rate for considering the environment solved at $92\%$. Means and $95\%$ confidence intervals are reported over 15 trials, 5 trials for the LM-Baseline.

## 5 DISCUSSION

In both experiments the GPT-4 agent attains a significantly higher success rate in fewer total samples than the baselines. Figure 3 shows the GPT-4 LLM Agent attains a $92\%$ success rate (matching the performance upper bound of the Oracle Agent) after only $600k$ environment steps, a small fraction of the steps of the baseline. The baseline agents are not able to generalize to all 162 tasks and only reaches a success rate of $\approx 80\%$ after 21 million steps. Note that while the WVF pretraining for our method requires 19 million steps, the pretraining objective does not include any language instructions and is distinct from the downstream task objective. The baseline learns the downstream tasks and language for its first 19 million training steps and still does not solve the 162 tasks. These results show the necessity of compositional representations for being able to learn large numbers of compositional tasks in a sample efficient manner.

Figure 4 demonstrates that the GPT-4 agent is able to generalize well to held-out tasks. The performance of the agent on training tasks and held-out task is very similar. This is expected given the

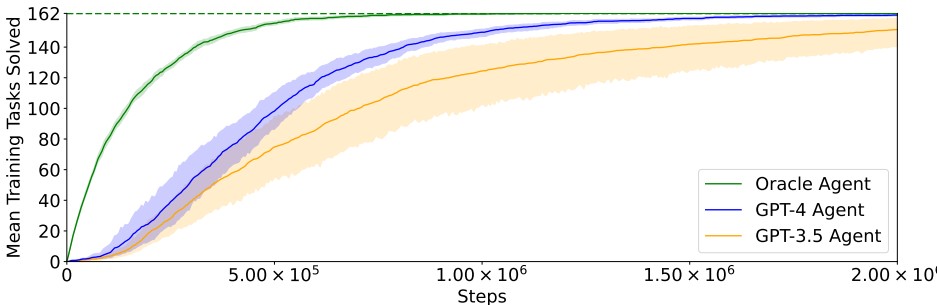

Figure 5: The mean number of tasks solved is plotted against the number of environment steps. This quantity is equal to the total number of in-context examples present in the LLM agents in-context example set at that step. Because the Oracle Agent has access to the ground truth Boolean expressions for each task, it solves tasks immediately. The population of tasks remains constant, so the number of unsolved tasks decreases over time, leading to a logistic learning curve for the LLM agents and an exponential decay in the rate at which new tasks are solved for the Oracle Agent. Means and 95% confidence intervals are reported over 10 trials.

ability to generalize compositionally in both the policy and language spaces. The LM-Baseline agent cannot generalize well between the training and held-out tasks as it lacks the necessary compositional representations. Note that this experiment trains the LLM and LM-Baseline agent for fewer steps than the 162 task experiment: one million and 21 million steps respectively.

In the 162 task learning experiment, the GPT-3.5 agent does not exceed the performance of the baselines even after two million steps indicating poor generalization. Confirming this, Figure 5 shows the mean number of tasks solved, which is the same as the number of total number of potential in-context examples that can be selected from during inference. Despite the GPT-3.5 agent solving most of the tasks, this does not transfer to a high evaluation success rate in Figure 3. The variance of the GPT 3.5 agent is also higher than the GPT-4 agent. This is caused by relatively worse generalization from available in-context examples than GPT-4. Highlighting the interpretability of our language-RL learning framework, we provide a qualitative analysis of the cause of the GPT-3.5 agent's lower performance in Appendix Table 3.

## 6 RELATED WORK

Our work is situated within the paradigm of RL, where novel tasks are specified using language and the agent is required to solve the task in the fewest possible steps. BabyAI (Chevalier-Boisvert et al., 2019) explores a large number of language-RL tasks, however it learns far fewer tasks simultaneously and their tasks do not involve negation. Another compositional RL benchmark CompoSuite (men, 2022) does not include language, and has fewer tasks than our 162 task benchmark when accounting for the number of unique goal conditions that could be specified in language.

Previous approaches have solved this problem using end-to-end architectures that are learned or improved using RL and a set of demonstrations (Anderson et al., 2018; Blukis et al., 2020; Chaplot et al., 2018). A problem with such approaches is a lack of compositionality in the learned representations. For example, learning to navigate to a red ball provides no information to the agent for the task of navigating to a blue ball. Moreover, demonstrations are hard to collect especially when users cannot perform the desired behavior. Some approaches demonstrate compositionality by mapping to a symbolic representation and then planning over the symbols (Dzifcak et al., 2009; Williams et al., 2018; Gopalan et al., 2018). However, these works do not learn these symbols or the policies to solve the tasks.

Compositional representation learning has been demonstrated in the computer vision and language processing tasks using Neural Module Networks (NMN) (Andreas et al., 2016; Hu et al., 2018), but we explicitly desire compositional representations both for the RL policies and the language command. Kuo et al. (2021) demonstrate compositional representations for policies, but they depend on a pre-trained parser and demonstrations to learn this representation. On the other hand, we use

large language models (Raffel et al., 2020) and compositional policy representations to demonstrate compositionality in our representations and the ability to solve novel unseen instruction combinations.

Compositional policy representations have been developed using value function compositions, as first demonstrated by Todorov (2007) using the linearly solvable MDP framework. Moreover, zero-shot disjunction (Van Niekerk et al., 2019) and approximate conjunction (Haarnoja et al., 2018; Van Niekerk et al., 2019; Hunt et al., 2019) have been shown using compositional value functions. Nangue Tasse et al. (2020) demonstrate zero-shot optimal composition for all three logical operators—disjunction, conjunction, and negation—in the stochastic shortest path problems. These composed value functions are interpretable because we can inspect intermediate Boolean expressions that specify their composition. Our approach extends ideas from Nangue Tasse et al. (2020) to solve novel commands specified using language.

Recent works like *SayCan* use language models and pretrained language-conditioned value functions to solve language specified tasks using few-shot and zero-shot learning (Ahn et al., 2023). Shridhar et al. (2021) use pretrained image-text representations to perform robotic pick-and-place tasks. Other work incorporates learning from demonstration and language with large-scale pretraining to solve robotics tasks (Driess et al., 2023; Brohan et al., 2022). However, these works use learning from demonstration as opposed to RL. Furthermore, these approaches do not support negations of pretrained value functions that our method allows. More importantly, their methodology is unsuitable for continual learning settings where both the RL value functions and language embeddings are improved over time as novel tasks are introduced.

Shin et al. (2021) utilize LLMs to learn semantic parsers using few-shot learning with in-context examples and Schick et al. (2023) uses an LLM to learn a semantic parser in a weakly supervised setting. Our method is distinct as we use policy rollouts in an environment as the supervision with in-context learning.

# 7 LIMITATIONS AND FUTURE WORK

One limitation of our method is the need for a pretraining phase where a curriculum is required to learn the basis set of WVFs. In future work, we plan on addressing this through experiments that simultaneously learn both the underlying WVFs and the language-instruction semantic parser using only RL on randomly selected tasks. This is a challenging optimization problem as the WVF models and the semantic parser must be optimized simultaneously to ensure that the WVFs form a good basis for the space of language tasks.

Our future work will also investigate our method's performance in simulated and real-world compositional RL tasks including vision and language navigation (VLN) and robotic pick-and-place tasks. The current environment has a discretized action space (although it utilizes images for state information); while this might limit the method's applicability to some real-world RL tasks, both VLN and pick-and-place tasks have been pursued in discretized forms (Anderson et al., 2018; Zeng et al., 2020). Both of these tasks could benefit from our method, as they require solving goal-reaching tasks which often have compositional language and task attributes. As one example, pick-and-place tasks are often compositional in terms of object type and locations for placing objects.

# 8 CONCLUSION

We introduced a method that integrates pretraining of compositional value functions with large language models to solve language tasks using RL. Our method rapidly solves a large space of RL tasks specified in language completely. Demonstrating efficacy across 162 tasks with reduced sample requirements, our findings also further differentiate the capabilities of GPT-4 from its predecessor, GPT-3.5, in semantic parsing using a RL signal. The amalgamation of compositional RL with language models provides a robust framework for reducing the sample complexity of learning RL tasks specified in language.

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

## A   APPENDIX

Table 3 shows the in-context examples that the GPT-3.5 agent has accumulated at the end of 2 million steps of learning the 162 tasks. As shown many of these expressions are not consistent or needlessly complicated. This helps to explain the relatively poorer performance of the GPT-3.5 agent which produces much noisier semantic parses and thus has much higher variance and lower performance than the GPT-4 agent. Reducing the sampling temperature during learning leads to better expressions, but at the cost of slower exploration and learning.

Table 3: In-context examples for the GPT-3.5 agent at the end of 2 million attributes.

| Language Instruction | Boolean Expression |
|---|---|
| pick up a ball or a grey object | $Symbol\_0 \mid Symbol\_4$ |
| pick up a box that is not grey | $\sim Symbol\_4 \& Symbol\_2$ |
| pick up a grey ball | $Symbol\_0 \& Symbol\_4$ |
| pick up a ball or an object that is not grey | $(Symbol\_0 \mid Symbol\_1) \mid \sim Symbol\_4$ |
| pick up a ball that is not grey | $Symbol\_0 \& \sim Symbol\_4$ |
| pick up a grey object or not a ball | $(Symbol\_0 \& Symbol\_4) \mid \sim Symbol\_0$ |
| pick up a grey object that is not a ball | $\sim Symbol\_0 \& Symbol\_4$ |
| pick up a grey object or not a key | $(Symbol\_0 \& Symbol\_1) \mid (\sim Symbol\_2 \& Symbol\_4) \mid \sim Symbol\_5$ |
| pick up an object that is not grey | $(Symbol\_0 \& Symbol\_1) \mid \sim Symbol\_4$ |
| pick up an object that is not grey and not a ball | $\sim Symbol\_4 \& \sim Symbol\_0$ |

Table 4: Task attributes and Boolean grammar. The symbols uniquely identify each learned WVF.

| Task Attributes | | Boolean Grammar | |
|---|---|---|---|
| **Colors** | **Objects** | **Symbols** | **Operators** |
| $red$, $purple$, $grey$ $green$, $yellow$, $blue$ | $key$, $ball$, $box$ | $Symbol\_0, Symbol\_1,\ldots,Symbol\_8$ | AND: & OR: |, NOT: $\sim$ |

Table 5: Hyperparameters for the LLM Agent.

| LLM Agent Hyperparameters | |
| --- | --- |
| LLM | GPT-4 and GPT 3.5 |
| Beam Width | 10 |
| Rollouts | 100 episodes |
| In-Context Examples | 10 |
| Training Temperature | 1.0 |
| Evaluation Temperature | 0.0 |

Table 6: Hyperparameters for world value function pretraining. The Adam optimizer was introduced by Kingma & Ba (2015).

| WVF Learning Hyperparameters | |
| --- | --- |
| Optimizer | Adam |
| Learning rate | 1e-4 |
| Batch Size | 32 |
| Replay Buffer Size | 1e3 |
| $\epsilon$ init | 0.5 |
| $\epsilon$ final | 0.1 |
| $\epsilon$ decay steps | 1e6 |

