# OpenReview forum: "Compositional Instruction Following with Language Models and Reinforcement Learning"
_ICLR.cc/2024/Conference — Submitted to ICLR 2024_

### Official Review · Reviewer_6vbo · 2023-10-26

**Soundness:** 3 good
**Presentation:** 1 poor
**Contribution:** 2 fair
**Rating:** 5
**Confidence:** 4

**Summary:**

In this paper, the authors address the task of training reinforcement learning agents with goals defined by language instruction, focusing on reducing sample complexity and generalization. They propose to use LLM to parse the instructions to simple boolean expressions for World Value Functions (WVF) learning. WVF was introduced in a previous paper, utilizing logical operators to design value functions to solve compositional tasks. The approach is evaluated in BabyAI environments, achieving a  success rate of 92%, which matches the upper-bound performance of an Oracle parser, outperforming the standard baseline in sample efficiency and generalization.

**Strengths:**

- The paper effectively leverages LLM to enhance reinforcement learning within the BabyAI environments.
- It explores a substantial set of 162 tasks, showcasing the breadth of the research.
- The proposed approach delivers superior results.

**Weaknesses:**

- The approach is incremental.  Since components like WVFs were known before (Nangue Tasse et al., 2020), the main contribution is the usage of LLM. However, it is straightforward to apply LLM to parse the instruction especially since previous papers suggested LLM as a good parser (Shin et al., 2021).
- In terms of practicality, for BabyAI, it is easy to program a perfect parser, and thus, it is not necessary to use LLM in this task.
- The baseline is too primitive. It is not surprising that the proposed method outperforms the baseline because it has access to additional information (e.g. compositions of the instruction). Please consider other baselines on Baby AI such as those in Carta et al., (2023)
- The pre-training time can be an issue. In Figure 3, after the pertaining time of the proposed method, the baseline already converges to a reasonable performance. Although the baseline cannot improve further, its reasonable performance may outweigh the cost of additional training of the proposed method
- The writing is hard to follow. More background on WVF is needed to understand the approach. The abstract also lacks information to capture the main idea of the paper (use LLM to parse instructions for WVFs). The method combines different components from different papers, so it is better to present an algorithm to describe the whole workflow in detail (especially the part that uses parsed symbols to select the WVF basis and the training of WVFs in the downstream task).


Carta, Thomas, Clément Romac, Thomas Wolf, Sylvain Lamprier, Olivier Sigaud, and Pierre-Yves Oudeyer. "Grounding large language models in interactive environments with online reinforcement learning." arXiv preprint arXiv:2302.02662 (2023).

**Questions:**

- What is trained during the downstream tasks? Are you fine-tuning the basic value functions?
- How many runs are used to generate the plot in Figure 3? The variance of GPT 3.5 is high.
- Table 2, the first Assistant should not have ~?
- Can you summarize 18 goals in a table? This sentence is hard to understand "Each WVF is implemented using |G| = 18 CNN-DQN (Mnih et al., 2015) architectures"
- How did you generate 162 tasks? Are any criteria used?

---

> ### Author Response · Authors · 2023-11-16
>
> Thank you for your time and feedback on our work. We've addressed the comments in the order listed.
>
> 1. We utilize a learned semantic parser that uses in-context learning and feedback from the environment. To the best of our knowledge, both our semantic parsing method and the scale and diversity of simultaneously learned language-RL tasks is unique in the literature. It is not a priori certain that semantic parsing should work in the RL setting on this quantity of tasks. Our method learns the parser from environment feedback, a much weaker form of supervision than is typically assumed. Further, we use our method to solve language grounding tasks involving negation, which are under-explored (Pillai and Matuszek 2018).
>
> We evaluate our agent on a much larger set of simultaneous tasks than previous works using WVF, which instead evaluated on individual tasks with hand-specified compositions of value functions. These works assumed ground-truth Boolean expressions, which is a very strong assumption. Our method relaxes this assumption and instead learns these Boolean expressions from environment feedback which is non-trivial.
>
> Our work does build on pre-existing methods. But many related works contain applications of existing methods, including SayCan (Ahn et al. 2022) which combines skill learning from demonstration with few-shot LLM planning, PaLM-E (Driess et al. 2023) which combines pre-existing vision and language representations to solve vision-language tasks. But we also go beyond the existing literature and demonstrate the utility of a novel semantic parsing method for RL settings and the advantage of compositionality at both the RL and language levels for solving a large set of language-RL tasks.
>
> 2. The original BabyAI domain is intentionally structured such that a perfect parser for language instructions can be defined, and this makes it the right domain for developing and evaluating our method. Because we can define a parser, we are able to benchmark our method against an Oracle and validate our method by showing that our method converges to the Oracle’s performance. Many recent works published at top conferences have utilized BabyAI for evaluation for similar reasons including Carta et al. (2022) EAGER: Asking and Answering Questions for Automatic Reward Shaping in Language-guided RL and Li et al. (2022), Pre-Trained Language Models for Interactive Decision-Making. In future work we plan on generalizing our method to other domains.
>
> 3. We appreciate your feedback on our baseline. We are updating the baseline to utilize pretrained language representations: the SOTA all-mpnet-base-v2 sentence embedding model available in the SentenceTransformers library (Reimers and Gurevych 2019). Initial results show that this agent still requires more steps than our agent and does not generalize well to held-out tasks. We plan to share updated baseline results by the end of the discussion period. The similar performance of the new baseline is expected, as neural networks struggle with compositional generalization and to generalize systematically in this domain requires both compositional RL representations and language representations. While the new baseline agent may generalize better between language commands, it must still learn each task value function without the aid of compositional structure. To solve this many compositional tasks in a generalizable way requires compositional generalization at both the language and policy level, which our method enables.
>
> Carta et al. (2023) does not appear to be an appropriate baseline as they only evaluate in a custom text-world environment with a significantly simpler observation space. Our method utilizes value function approximation from images and text, providing greater applicability and we evaluate in a more challenging observation space.
>
> 4. We demonstrate that our method attains a higher success rate in fewer timesteps than the non-compositional baseline and also offers a simple means of interpretability which black-box methods like joint text-image value function learning do not. We utilize this interpretability to analyze the failure mode of the ablation using the less capable GPT-3.5 Agent. In terms of performance, our method significantly reduces (by more than 50%) the error rate over the baseline for an equal number of environment steps at convergence. Our method also generalizes better to held-out tasks as shown in Figure 4.
>
> 5. We appreciate your feedback and will improve the explanation of the WVF learning for a camera-ready submission and include additional details in the appendix.

---

> > ### Author Response · Authors · 2023-11-16
> >
> > Questions:
> > 1. During training on downstream tasks the WVFs are frozen and only the LLM agent is being updated through in-context learning.
> > 2. We utilized 10 trials in all experiments in Figure 3. GPT 3.5 has lower performance and more variance than the other methods because it learns a weaker semantic parser.
> > 3. Thank you for pointing out this typo, it will be corrected.
> > 4. The goal space is the cross product of the colors and object types summarized in Appendix Table 4. We will add a new table in the appendix to make this explicit.
> > 5. The 162 tasks are composed from the object types, colors, and task types in Table 1. We generate tasks from Boolean expressions over the task attributes, including negation of attributes. The background section (2. BabyAI Domain) provides an overview of our task space. We are happy to update this section for clarity based on your feedback.
> >
> >
> > References:
> >
> > Ahn, Michael, et al. "Do as I Can, Not as I Say: Grounding Language in Robotic Affordances." CoRL (2022).
> >
> > Driess, Danny, et al. "Palm-e: An embodied multimodal language model." ICML (2023).
> >
> > Shridhar, Mohit, Lucas Manuelli, and Dieter Fox. "Cliport: What and where pathways for robotic manipulation." Conference on Robot Learning. PMLR, 2022.
> >
> > Carta, Thomas, et al. "Eager: Asking and answering questions for automatic reward shaping in language-guided rl." Advances in Neural Information Processing Systems 35 (2022): 12478-12490.
> >
> > Li, Shuang, et al. "Pre-trained language models for interactive decision-making." Advances in Neural Information Processing Systems 35 (2022): 31199-31212.
> >
> > Carta, Thomas, et al. "Grounding large language models in interactive environments with online reinforcement learning." arXiv preprint arXiv:2302.02662 (2023).
> >
> > Nils Reimers and Iryna Gurevych. Sentence-bert: Sentence embeddings using siamese bert- networks. In Proceedings of the 2019 Conference on Empirical Methods in Natural Language Processing. Association for Computational Linguistics, 11 2019. URL https://arxiv. org/abs/1908.10084.
> >
> > Pillai, N., & Matuszek, C. (2018). Unsupervised Selection of Negative Examples for Grounded Language Learning. Proceedings of the AAAI Conference on Artificial Intelligence, 32(1). https://doi.org/10.1609/aaai.v32i1.12108

---

> ### Author Response · Authors · 2023-11-22
>
> Thank you again for your feedback and questions. We have updated Figures 3 and 4 with the results of using pretrained language representations in the baseline denoted "LM-Baseline". These new results do not show significant learning differences when using only pretrained language representations. As expected the agent cannot generalize compositionally without access to representations that allow for such generalization for both language and RL.
>
> In Figure 4, we have also run the baseline for more steps (21 million).

---

### Official Review · Reviewer_VNmK · 2023-10-28

**Soundness:** 3 good
**Presentation:** 3 good
**Contribution:** 2 fair
**Rating:** 5
**Confidence:** 4

**Summary:**

This paper proposes a framework for solving logical compositional goal-reaching tasks instructed by language. The framework is a two-stage approach. In the first stage, it pretrains a collection of value functions for basic tasks that can later be composed to build other tasks with reinforcement learning. In the second stage, it leverages a large language model to convert language instructions into logical expressions over the basic symbols obtained in the first stage. The predicted logical algebra is executed in the environment to obtain the success or not signal, and the successful examples are then added to the in-context examples to prompt the LLM. Empirical studies are conducted in the BabyAI environment focusing on different combinations of attributes. The proposed framework shows better overall success rates and better generalization ability than the baseline without compositional representation and LLM.

**Strengths:**

1. Combining the ability of zero-shot compositional generalization in boolean expressions of world value functions and parsing language instructions into logical expressions is a novel solution.
2. The generalization performance of the proposed method is impressive.
3. The idea of using environmental rollout signals as in-context examples as feedback to the LLM is interesting, and could be of benefit to readers who are integrating combining LLM with RL.

**Weaknesses:**

1. More empirical baselines should be studied to verify the effectiveness of the proposed method. For example, (a) defining some kind of action space for boolean expressions and then using RL to learn a policy that converts language instructions into boolean expressions. This baseline could better illustrate the contribution of LLM; (b) Fine-tuning the language model using the environmental feedback so as to show the impact of in-context learning from environmental feedback.
2. I think the empirical validation is not sufficient enough to support the claim made fully. The method purposefully treats all the basic value functions as symbols without semantic meaning so as to be applicable to different domains with different definitions of such value functions. Currently, it is only experimented on BabyAI, a relatively neat and simple environment. It would be better to run the method in some other domains (ideally more realistic domains, such as robot manipulation).
3. As the authors have mentioned, the proposed method requires pretraining a set of WVFs as a basis for further language instruction parsing. Currently, what the set of WVFs is composed of requires an in-depth understanding of the domain. Automating the design of WVFs or allowing to dynamically modify WVFs will strengthen this work.

**Questions:**

In this work, only the boolean expressions that lead to sufficiently high success rates are added to in-context samples. Does it mean that the correct boolean expression should be easy to explore for the LLMs? Is it possible to obtain more supervision from failed environmental rollouts?

---

> ### Author Response · Authors · 2023-11-16
>
> Thank you for your time and feedback on our work. We've addressed the comments in the order listed.
>
> 1. We appreciate your suggestions. We plan on including those suggested ablations in a camera-ready submission: a fine-tuned LLM baseline and a RL-based semantic parser that doesn’t utilize pre-trained LLM representations. In our experience we have found GPT-4 to be significantly better than other models for semantic parsing. We note that to our knowledge, our method is the first to learn semantic parsing using in-context learning with feedback from policy rollouts. This is a much weaker form of supervision than is typically assumed in semantic parsing. We believe this is an important contribution to existing work that maps language to structured representations for language instruction following tasks.
>
> 2. We believe BabyAI is the most appropriate domain for our work. It provides a scalable RL learning environment for a suite of compositional tasks. BabyAI is also easily extensible, allowing us to augment the domain with the largest set of simultaneously learned language-RL tasks (to the best of our knowledge). Other recent related works published at top conferences also select BabyAI for evaluation for similar reasons including (Carta et al. 2022 EAGER: Asking and Answering Questions for Automatic Reward Shaping in Language-guided RL), (Li et al. 2022, Pre-Trained Language Models for Interactive Decision-Making). The newer Carta et al. 2023 also use BabyAI, but they convert BabyAI to a simpler text-world domain (Grounding Large Language Models in Interactive Environments with Online Reinforcement Learning).
>
> 3. We appreciate your comment and plan on investigating learning of WVFs without a curriculum in future work.
>
> Questions:
> 1. Our approach builds on similar in-context semantic parsing work like Toolformer, but goes beyond these methods as it is the first to use in-context learning from environment feedback. We have not been able to devise an effective method where the agent utilizes negative examples for in-context learning. We can however utilize negative samples in the fine-tuning setting through unlikelihood training (Welleck et al. 2019).
>
> References:
>
> Carta, Thomas, et al. "Eager: Asking and answering questions for automatic reward shaping in language-guided rl." Advances in Neural Information Processing Systems 35 (2022): 12478-12490.
>
> Li, Shuang, et al. "Pre-trained language models for interactive decision-making." Advances in Neural Information Processing Systems 35 (2022): 31199-31212.
>
> Carta, Thomas, et al. "Grounding large language models in interactive environments with online reinforcement learning." arXiv preprint arXiv:2302.02662 (2023).
>
> Welleck, Sean, et al. "Neural text generation with unlikelihood training." arXiv preprint arXiv:1908.04319 (2019).

---

> > ### Comment · Reviewer_VNmK · 2023-11-17
> >
> > Dear authors,
> >
> > Thank you for your response. I would like to clarify weakness 2 in my review. I am asking for applications in more domains mainly because more evidence is required to support the claim that treating all WVFs as symbols allows “the method to be applied to any basis of tasks” (In the beginning of Sec. 3). I agree that BabyAI is an appropriate testbed for your method but still curious to see if it could be applied in other domains/with other basis of tasks.

---

> > > ### Author Response · Authors · 2023-11-21
> > >
> > > Thank you for your question. Our method does not require access to the semantics of the base tasks, as has been assumed in related work. This means that in principle it can be applied to any basis of tasks without this extra human-engineered information. We plan to clarify this distinction in section 3 for a camera-ready version of the paper. We can also perform an additional evaluation for another domain to demonstrate this point.
> > >
> > > Prior works have also applied WVF to other domains and task bases without language. These include a 3D physics based environment “Bullet-Safety-Gym” in “Safety-Aware Task Composition for Discrete and Continuous Reinforcement Learning” (Leahy et al. 2023) and other domains in “Skill Machines: Temporal Logic Composition in Reinforcement Learning” (Tasse et al. 2022). We believe this demonstrates the applicability of the WVF method to other domains.
> > >
> > > Leahy, Kevin, Makai Mann, and Zachary Serlin. "Safety-Aware Task Composition for Discrete and Continuous Reinforcement Learning." arXiv preprint arXiv:2306.17033 (2023).
> > >
> > > Tasse, Geraud Nangue, et al. "Skill machines: Temporal logic composition in reinforcement learning." arXiv preprint arXiv:2205.12532 (2022).

---

> > > ### Author Response · Authors · 2023-11-22
> > >
> > > Thank you again for your feedback and questions. We have updated Figures 3 and 4 with the results of using pretrained language representations in the baseline denoted "LM-Baseline". These new results do not show significant learning differences when using only pretrained language representations. As expected the agent cannot generalize compositionally without access to representations that allow for such generalization for both language and RL.
> > >
> > > In Figure 4, we have also run the baseline for more steps (21 million).

---

### Official Review · Reviewer_wEbe · 2023-10-30

**Soundness:** 2 fair
**Presentation:** 3 good
**Contribution:** 2 fair
**Rating:** 3
**Confidence:** 4

**Summary:**

This paper proposes a method for performing RL tasks specified using natural language commands. The approach involves using an LLM to map a given natural language specification to an expression representing a Boolean combination of primitive tasks and then applying an existing compositional RL technique for obtaining an optimal policy/Q-function for the corresponding task. LLM prompts are generated automatically including in-context examples which are obtained from prior interactions with the environment. Experiments indicate that this approach is more sample efficient in learning to perform multiple tasks (specified in natural language) when compared to a baseline that trains a policy end-to-end. The authors also show that their approach improves the transferability of the learnt policy to unseen tasks.

**Strengths:**

- Although the idea of generating a structured representation from natural language is not new (e.g., code generation), the idea of using feedback from the environment to create in-context examples and the idea of using LLM generated logical specifications in conjunction with compositional RL approaches are interesting.
- The paper is well-written and fairly easy to follow. The problem is well motivated and the solution is well explained.
- The approach seems to enable learning policies that can follow a wide range of natural language instructions.

**Weaknesses:**

- _Need for user-defined primitive tasks:_ The compositional approach presented in this paper requires manual specification of primitive tasks and training a Q-function for each primitive task. This set of primitive tasks might not be readily available; which makes this approach not easily applicable to realistic scenarios. Furthermore, the baseline approach does not have access to any such additional information.

- _Unfair comparison to baseline:_ As mentioned above, the baseline does not have access to additional information that is being used to achieve the decomposition necessary for the approach presented in this paper. Furthermore, the baseline approach does not have access to any pre-trained LLM and has to learn the logical structure in the natural language instructions from scratch by interacting with the environment. It is worth considering a baseline that uses the output of a hidden layer of a pre-trained LLM as the encoding of the natural language specification.

- _Only Boolean combinations supported:_ The instructions are assumed to always correspond to a Boolean combination of primitive tasks. There is no support for temporal compositions such as doing one task after another. Furthermore, this reduces the problem to finding the mapping between keywords in the textual task description and the set of symbols used to denote the primitive tasks. In fact, I believe LLMs such as GPT3.5 can already construct Boolean formulas from such language instructions if this mapping is given. Although this mapping is being learned in the proposed approach, I am not sure if there are use cases where the set of primitive tasks is available but the mapping to the symbols is not.

**Questions:**

1. Did you consider using a pre-trained encoding of the natural language instruction instead of learning this from scratch in the baseline? Would the peak performance of the baseline still remain lower than the GPT4.0 LLM agent?
1. How would an LLM agent that has mappings from primitive tasks to symbols representing them (but has to still generate the Boolean formula) perform in comparison to the proposed approach (and the oracle agent)?
1. Do you have any thoughts on how your approach can be modified to support a richer set of tasks (for example, involving temporal compositions)?

---

> ### Author Response · Authors · 2023-11-16
>
> Thank you for your time and feedback on our work. We've addressed the comments in the order listed.
>
> Need for user-defined primitive tasks:
> Current robotics deployments require pre-defined primitive tasks and for human design of the low-level controllers. For example, SOTA language instruction following methods like SayCan (Ahn et al. 2022) rely on extensive human-design of both high level tasks and low-level control through manual demonstrations. This work is among the first to relax these requirements (the low-level controllers are learned) while providing the ability to compose these controllers, and is the largest set of such tasks learned simultaneously with language and RL.
>
> Unfair comparison to baseline:
> We are updating the baseline to utilize pretrained language embeddings from the SOTA all-mpnet-base-v2 sentence embedding model from SentenceTransformers library (Reimers and Gurevych 2019) . Initial results show that this agent still requires more steps than our agent and does not generalize well to held-out tasks. We plan to share updated baseline results by the end of the discussion period. The similar performance of the new baseline is expected, as neural networks struggle with compositional generalization and to generalize systematically in this domain requires both compositional RL representations and language representations. While the new baseline agent may generalize better between language commands, it must still learn each task value function without the aid of compositional structure. To solve this many compositional tasks in a generalizable way requires compositional generalization at both the language and policy level, which our method enables.
>
> Only Boolean combinations supported:
> Prior work in semantic parsing with LLM: to our knowledge, our method is the first to learn semantic parsing using in-context learning with feedback from environment feedback. This is a much weaker form of supervision than is typically assumed in semantic parsing and represents an important contribution to existing language instruction following literature that maps natural language to formal representations.
>
> Even for cases where the mapping from task primitives to symbols is known, one challenge is that language is ambiguous and there is a many-to-one mapping of language instructions to individual tasks, and language symbols to task primitives. Indeed knowing only the semantics of the symbols is insufficient in our domain for learning to form valid Boolean expressions as our tasks involve combinations of primitive task negation, intersection, and conjunction. Lastly, Boolean combinations are able to express many interesting tasks. Because we support Boolean task specifications, our method also naturally generalizes to task negation. Negation is understudied in the grounded language literature (Pillai and Matuszek 2018) and to the best of our knowledge our paper also solves the largest and most diverse set of language-negation RL tasks. We leave temporal compositions of tasks to future work.
>
> Questions:
> 1. Your question is well taken, and we are updating the baseline to utilize pretrained language embeddings as mentioned above.
> 2. We plan on adding ablations where the agent has access to the underlying semantics of the WVFs in a camera ready submission. Previous exploration has shown that this improves the sample efficiency of the agent.
> 3. We believe that temporal compositions are useful, but fall outside of the scope of this work. Instead we evaluate a task-space based on Boolean expressions which can express many useful types of tasks, for example “serve breakfast with coffee and no milk.”
>
> References:
>
> Ahn, Michael, et al. "Do as I Can, Not as I Say: Grounding Language in Robotic Affordances." CoRL (2022).
>
> Pillai, N., & Matuszek, C. (2018). Unsupervised Selection of Negative Examples for Grounded Language Learning. Proceedings of the AAAI Conference on Artificial Intelligence, 32(1). https://doi.org/10.1609/aaai.v32i1.12108

---

> ### Author Response · Authors · 2023-11-22
>
> Thank you again for your feedback and questions. We have updated Figures 3 and 4 with the results of using pretrained language representations in the baseline denoted "LM-Baseline". These new results do not show significant learning differences when using only pretrained language representations. As expected the agent cannot generalize compositionally without access to representations that allow for such generalization for both language and RL.
>
> In Figure 4, we have also run the baseline for more steps (21 million).

---

> > ### Comment · Reviewer_wEbe · 2023-11-23
> > **Author Response Acknowledgement**
> >
> > I thank the authors for the detailed response. I appreciate the addition of the new baseline. However, I still have the other concerns mentioned in my review. Specifically, although natural language is ambiguous, LLMs can extract structure from natural language and the proposed approach is an illustration of the same which is not novel by itself. I believe the novelty comes from creating in-context examples which helps to learn a mapping from primitive tasks to their symbols. However, as I mentioned, the experiments involve scenarios where such a mapping is actually available to the user but is hidden from the agent.
> >
> > Overall, I am maintaining my score but encourage the authors to consider more interesting scenarios where the specific setup is useful.

---

> > > ### Author Response · Authors · 2023-11-23
> > >
> > > We thank you for your time but respectfully disagree with this criticism. We developed a novel method for semantic parsing that learns from environment feedback with in-context learning in an RL loop. Our work also evaluates on the largest available set of simultaneously learned language-RL tasks of any prior work. It is not obvious from prior work that this should succeed. Indeed we have tried using smaller and other language representations without success. In our view our experiments are sufficient to confirm the validity of our method and this method advances the state of the art in language-RL research.

---

### Official Review · Reviewer_Gnf9 · 2023-10-30

**Soundness:** 3 good
**Presentation:** 3 good
**Contribution:** 2 fair
**Rating:** 5
**Confidence:** 4

**Summary:**

The paper proposes to combine natural language instructions with boolean compositional value functions for improved generalization and sample efficiency of instruction following models. The method leverages an LLM to parse a natural language instruction into the corresponding boolean formula. Based on this boolean formula the action value function for the specific instruction is constructed following the framework of Nangue (2022). The method is evaluated on a set of 162 instruction following tasks build on top of the BabyAI environment. The experiments investigate the sample efficiency and generalization performance of their method against a non-compositional CNN-DQN baseline using either GPT3.5 or GPT4 as the underlying LLM. The method generalizes better to a held-out set of tasks. For all experiments the parsing of GPT3.5 leads to worse results than the parsing with GPT4.

**Strengths:**

The method and experiments were presented in a clear and understandable way, which was easy to follow.

The method shows improved generalization performance.

A reward signal is used to choose in-context learning examples for the LLM, which is to the knowledge of the authors as well as mine a novel approach of prodividing environment feedback to LLMs.

Using LLMs as semantic parsers to bridge the gap between natural language instructions and logical formulas tackles the problem of how to obtain logical formulas for certain tasks without much engineering effort. Prior work often assumes such formulas or a parser as given.

**Weaknesses:**

Sample Efficiency:

- Given the results in Figure 3. I do not think one can make the statement that the method is more sample efficient, than the baseline. The baseline stops improving after the number of environment steps has reached the number of pretraining steps. To me one method is more sample efficient than the other if it reaches the same performance with less samples. The conclusion that can be drawn from the plot is that the method outperforms the baseline in terms of final performance. And to me it is not immediately clear why the baseline is limited in its final performance, would a larger network or more carefully tuned hyperparameters lead to better final performance?

Baseline:

- Why were there no pretrained language embeddings used for the baseline? That should have a large impact on at least the sample efficiency of the baseline.

Formalities:

- I would present Figure 4 in the same way Figure 3 is presented. The authors state that because they look at generalization performance and not sample efficiency a clear separation of the pretraining samples and the training samples is not necessary. However, the plot looks at generalization performance after a certain number of environment steps, and for a given point on the x-axis the current plot compares the generalization performance of two method trained with different number of samples. If I understood it correctly the number of steps for which the baseline was trained in that scenario is 10 million, so it is close to only half of all the pretraining steps. Since the generalization performance of the baseline is so low I do not think it makes much difference in the conclusions that can be drawn, but would result in a more accurate comparison.

**Questions:**

- The sucess rate that boolean expressions need to satisfy to be added to the database of boolean expressions that can be added to the prompt, seems to be set by expected performance of the oracle parser. Did you perform any ablations on that hyperparameter? A lower threshold might lead to worse formulas being added but also more formulas early on and vice versa. It would be interesting to see how sample efficiency changes with that parameter.

- In the boolean expressions abstract symbols were used for concepts such as "grey object", why not the corresponding natural language labels. Would that not make parsing for the LLM easier?

- In paragraph 3.2.1 it says that during training a beam of 10 parses was looked at. During the evaluation of the agent, did the agent choose the most likely parsing and evaluate it for 100 episodes or was the average evaluation of all 10 parses looked at ?

---

> ### Author Response · Authors · 2023-11-16
>
> Thank you for your time and feedback on our work. We've addressed the comments in the order listed.
>
> Sample Efficiency:
> Your comment on the results of Figure 3 are well taken. We are claiming a sample efficiency gain and benchmark our model against the best performing non-compositional baseline DQN agent we could train. This baseline agent does continue to increase in performance, though at a decreasing rate of improvement. The x-axis scale change could also make this hard to see. Given that we are working in the function-approximation setting and learning value functions from text and images, the agent is not guaranteed to converge to the optimal action-value function. If we were evaluating in a tabular setting where convergence is guaranteed, it would be appropriate to run our baseline until convergence. Our setting is more challenging than previous similar compositional domains. To the best of our knowledge it contains the most simultaneously learned language-RL tasks. Since we have no guarantee of convergence, we instead run the baseline agent for many more steps than it takes for our method to converge to the Oracle agent’s performance. We thereby show that our method requires fewer steps than the baseline to reach the Oracle agent’s performance.
>
> Baseline:
> We are updating the baseline to utilize pretrained language embeddings from the SOTA all-mpnet-base-v2 sentence embedding model from SentenceTransformers library (Reimers and Gurevych 2019) . Initial results show that this agent still requires more steps than our agent and does not generalize well to held-out tasks. We plan to share updated baseline results by the end of the discussion period. The similar performance of the new baseline is expected, as neural networks struggle with compositional generalization and to generalize systematically in this domain requires both compositional RL representations and language representations. While the new baseline agent may generalize better between language commands, it must still learn each task value function without the aid of compositional structure. To solve this many compositional tasks in a generalizable way requires compositional generalization at both the language and policy level, which our method enables.
>
> Formalities:
> We will run the baseline for more steps for Figure 4 as suggested, to provide a more fair comparison of generalization performance at each timestep. We plan to update these results by the end of the discussion period.
>
>
> Questions:
> 1. Thank you for your suggestion. We arrived at that hyperparameter empirically but have not completed a full multi-trial evaluation of ablated values. We can include these in a camera-ready submission.
> 2. We believe this helps demonstrate the applicability of our method to situations where the semantics of the value functions are either not easily expressed in language or are otherwise challenging to enumerate.
> 3. During training each parse in the beam is evaluated for 100 episodes, until a high success rate parse is found, or all parses in the beam have been exhausted. The section will be updated in a camera-ready submission to make this clearer.
>
> References:
> Nils Reimers and Iryna Gurevych. Sentence-bert: Sentence embeddings using siamese bert- networks. In Proceedings of the 2019 Conference on Empirical Methods in Natural Language Processing. Association for Computational Linguistics, 11 2019. URL https://arxiv. org/abs/1908.10084.

---

> > ### Comment · Reviewer_Gnf9 · 2023-11-20
> >
> > Thank you for your replies to my comments.
> >
> > All my questions have been answered. One follow up question would be whether you could give an example where the symbols of the boolean expressions cannot be expressed with a natural language equivalent but instructions are still expressed in natural language ?
> >
> > I think adding the embeddings from the sentence embedding model and the change in the x-axis on the generalization performance plot are both improving the paper. I will wait for the updates.
> >
> > Regarding the comments on sample efficiency I do not see how the baseline keeps improving after the vertical line at $1.90 \times 10^7$. It seems to me that the performance is slightly above 0.8 at that point in training and at the end of training?

---

> > > ### Author Response · Authors · 2023-11-21
> > >
> > > Great - glad we could address your concerns
> > >
> > > Thank you for your follow up questions:
> > > 1. We plan on adding ablations where the agent has access to the underlying semantics of the WVFs in a camera ready submission. Previous exploration has shown that this improves the sample efficiency of the agent.
> > >
> > > However, our view is that there are cases when the semantics of the base skills are not easily captured or may be challenging to enumerate. If those language labels were required by a system, that would represent another component of hand-engineering. Such labeling of skills with natural language labels has been studied in other works like SayCan, but not needing those labels offers more flexibility.
> > >
> > > People can use a variety of language to refer to particular objects and actions and we want to be able to understand those mappings from interaction with the environment without explicit supervision.
> > >
> > > For example, imagine an agent that can serve various breakfast items. In our approach one might say “serve breakfast with coffee and cereal.” This could be expressed as the conjunction of  serve_coffee AND serve_cereal. But what if we want to learn a command “serve a low calorie breakfast”? This requires understanding of the properties of the primitive skills (the number of calories in the items) that goes beyond simple language labels, and it’s important for agents to be capable of learning in the absence of that human-engineered additional knowledge (which represents another assumption of language-RL domains).
> > >
> > > We want to be able to understand a wide range of potential mappings without explicit supervision for all of them.
> > >
> > > 2. We apologize if the change in scale makes the graph more challenging to interpret. We intended to point out that the trend line visible to the left of the vertical line indicates continued improvement of the baseline model. This trend will necessarily be less visible over the shorter interval (many fewer timesteps) to the right of this vertical line, and as the rate of improvement is decreasing, noise from the randomness of evaluation will begin to dominate any improvement over short intervals. Therefore it may look flat, but is still likely to be slightly increasing.

---

> > > > ### Author Response · Authors · 2023-11-22
> > > >
> > > > Thank you again for your feedback and questions. We have updated Figures 3 and 4 with the results of using pretrained language representations in the baseline denoted "LM-Baseline". These new results do not show significant learning differences when using only pretrained language representations. As expected the agent cannot generalize compositionally without access to representations that allow for such generalization for both language and RL.
> > > >
> > > > In Figure 4, we have also run the baseline for more steps (21 million) as suggested.

---

> > > > > ### Comment · Reviewer_Gnf9 · 2023-11-23
> > > > >
> > > > > Thank you for the update.
> > > > >
> > > > > I think additional experiments strengthen the paper. However, I am still not convinced by the sample efficiency claims, so I will keep my score. I think two avenues worth exploring are:
> > > > >
> > > > > - Reducing the sample complexity of pretraining.
> > > > >
> > > > > - As indicated above using natural language semantics for the symbols might improve sample efficiency of the method further.

---

### Meta-Review · Area_Chair_WgjR · 2023-12-09

**Metareview:**

This paper proposes a framework for language-based solving of logical goal-reaching tasks. Experiments are conducted in the BabyAI environment. However, several concerns have been raised with limited baselines used in comparison, limited environment complexity as BabyAI is quite simple, and the overall method being incremental given that WVFs is a known technique and LLM parsing is a straightforward idea. After the rebuttal, the reviewers are not sufficiently convinced on the importance of the proposed work.

**Justification For Why Not Higher Score:**

Limited environment settings and baselines. Limited novelty of proposed ideas.

**Justification For Why Not Lower Score:**

N/A

---

### Decision · Program_Chairs · 2024-01-16

Reject